

# Influence of Floodplains and Groundwater Dynamics on the Present-Day Climate simulated by the CNRM Model

Bertrand Decharme[1] and Jeanne Colin[1]

[1]Centre National de Recherches Météorologiques (CNRM), Météo-France, CNRS, Université de Toulouse, Toulouse, France

**Correspondence:** Bertrand Decharme (bertrand.decharme@meteo.fr)

**Abstract.** The climate impacts of floodwater stored over large inundated areas and groundwater stored in large unconfined aquifers at the global-scale are not yet well-documented, despite their potential to affect the atmosphere through contributions to land surface evapotranspiration fluxes. To address these gaps in knowledge, the present study aims to assess the potential role of these processes on present-day climate using the CNRM-CM6-1 global climate model, the physical core of the Earth System Model (ESM) used by the French National Center for Meteorological Research for climate projections. This model includes a dynamic river flooding scheme and a groundwater scheme accounting for the 218 world's largest unconfined aquifer basins. The study consists of four experiments, each with five ensemble members driven by observed monthly sea surface temperature and sea ice cover for the 1980-2014 period. The experiments include configuration variations where both groundwater and floodplain processes were activated or deactivated, as well as configurations where each process was individually activated. The various forcings used in CNRM-CM6-1 adhere to the CMIP6 recommendations. The False Detection Rate test is employed to assess the significance of field differences. This study found that the impact of groundwater and floodplains on precipitation and 2-meter air temperature biases is predominantly positive in comparison to observations, although on a very regional scale. Additionally, the model's ability to simulate terrestrial water storage and river discharges is enhanced by these processes. The improvement in land hydrology with floodplains and groundwater is attributed to their ability to increase the hydrological memory of the model. Overall, the study highlights the importance of incorporating groundwater and floodplain processes into ESMs to improve the understanding of land surface-atmosphere interactions and the accuracy of climate simulations.



## 1 Introduction

Water has a special place in the Earth system simply because it is the main and original source of all life. Salt water, mainly contained in the oceans, makes up 97.5% of the volume of all water on Earth while only 0.001% is in the atmosphere (Horn-
berger et al., 1998). Inland freshwater accounts for only 2.5% of the continental water, of which 1.74% and 0.75% is contained in glaciers and aquifers. The high-water retention capacity of these two reservoirs therefore constitutes a kind of force of inertia in front of rapid climate variations. Conversely, lakes, rivers, seasonal floodplains and soil moisture – 0.008% of the water residing on Earth – generally show a high temporal variability due to their relatively low storage capacity. Subject to climatic hazards, the rapid evolution of these surface reservoirs can have dramatic consequences for populations through floods or
droughts (Di Baldassarre et al., 2017). In turn, these reservoirs will influence the climate through their control on plant life and land surface water, energy and carbon cycles (Seneviratne et al., 2010; Saunois et al., 2020; Canadell et al., 2021). The study of these interactions at the global-scale requires Ocean-Atmosphere-Global-Circulation Models (OAGCMs) that have been developed focusing on the physics and dynamics of the atmosphere, oceans (including sea ice), and the hydrometeorology of the land surface (Bonan and Doney, 2018). These OAGCMs are the core physical component of Earth System Models (ESM),
which also represents the biogeochemical processes involved in the atmospheric chemistry and the carbon and nitrogen cycles. In these climate models, the land surface processes are simulated using Land Surface Models (LSM) and/or lake models that provide realistic boundary conditions for the atmosphere in term of momentum, moisture, temperature, and energy, as well as River Routing Models (RRM) that simulate river discharges into the ocean – and potentially other reservoirs such as aquifers or floodplains – allowing closure of the global water budget.

The study of hydrological land surface processes plays an increasingly important role in the understanding of the climate system, its evolution, and its predictability. The global impact of soil moisture on climate – and inversely – is the largest-documented physical feedback between land surface and climate to date. Indeed, soil moisture controls the exchange of water and energy between the continents and the atmosphere through its direct influence on soil temperature and on land surface evapotranspiration (Seneviratne et al., 2010). By making things simpler, evaporation from bare soil is directly related to the
evolution of near-surface soil moisture and becomes potential when this moisture becomes very high, i.e. above a certain threshold generally taken to be equal to the field capacity (Mahfouf and Noilhan, 1991). The same applies to vegetation transpiration, however it stops when the root zone soil moisture falls below the wilting point of the plants. As a result, in transition zones between dry and wet climates, there is a significant coupling between this soil moisture and land surface evapotranspiration that will generate interactions and feedbacks with precipitation and air temperature (Douville et al., 2002;
Koster et al., 2004, 2006; Seneviratne et al., 2006, 2010, 2013). All this also implies that soil moisture is today considered as a significant source of predictability for monthly to seasonal climate forecasts (Dirmeyer et al., 2019). Spatial redistribution of soil moisture – usually related to topography (Beven and Kirkby, 1979) – also allows rainfall to be distributed between soil infiltration and runoff (Dunne and Black, 1970; Horton, 1933), affecting the water stock available for plants, groundwater recharge, seasonal flooding and river flow. The impact of lake or wetlands on climate is also well documented. Several studies
have shown that inland water like lakes or wetlands, especially those in the northern hemisphere, can have a significant influence



on the atmosphere by humidifying and cooling its lower layers (Bonan, 1995; Lofgren et al., 2002; Krinner, 2003; Balsamo et al., 2012; Le Moigne et al., 2016; Arboleda Obando et al., 2022). These impacts remain mostly local and confined to the lowest levels of the atmosphere. They are localized over the northwest Canada, the Great Lakes region of North America, the Scandinavia or east Africa. In these regions, these water bodies increase surface latent heat flux to the detriment of surface
sensible heat flux, injecting more humidity in the atmospheric boundary layer compared to vegetated or bare ground land surface.

The climate impacts of groundwaters stored in large unconfined aquifers at the global-scale are however not yet enough documented while it acts as a lower boundary for the overlaying unsaturated soil through upward capillarity rise. Indeed, groundwaters could affect the atmosphere – especially precipitation, temperature and humidity – through their contributions
to surface and root zone soil moisture and then to land surface evapotranspiration fluxes (Maxwell et al., 2007; Kollet and Maxwell, 2008; Maxwell et al., 2011; Vergnes et al., 2014; Maxwell and Condon, 2016; Decharme et al., 2019). Previous studies have started to explore this issue, but they remain mostly regional and over rather short periods of time (Anyah et al., 2008; Maxwell and Kollet, 2008; Ferguson and Maxwell, 2010; Keune et al., 2016; Larsen et al., 2016; Poshyvailo-Strube et al., 2024). To our knowledge, only two global-scale studies have been conducted, either using an idealized – and therefore
unrealistic – coupled land–atmosphere models with a prescribed globally homogeneous shallow water table depth (Wang et al., 2018), and secondly, using an empirical representation of hillslope flow along topography that allows soil moisture to converge to a fixed "lowland" fraction of a grid cell, neglecting hydrogeological groundwater processes (Arboleda Obando et al., 2022). At the continental scale, groundwaters could also affect the long-term hydraulic memory of the land surface through theirs capability to store water for significantly longer periods than in shallow unsaturated soils (Opie et al., 2020; Mu et al., 2021).
Another completely undocumented interaction at the global-scale is about the rule of floodwater stored over large inundated areas that take place after each rainy season along rivers especially over the tropics and northern latitudes (Lehner and Döll, 2004; Prigent et al., 2007; Yamazaki et al., 2011, 2015; Decharme et al., 2012). As for others inland water bodies, these seasonal floodplains could affect the overlying atmosphere through their relatively high evapotranspiration, which could enhance latent versus sensible heat exchange with the atmosphere.

To our knowledge, river overflow floodplains are ignored in all ESMs today while very few models attempt to simulate groundwater processes at the global-scale (Golaz et al., 2019; Danabasoglu et al., 2020; Seland et al., 2020), their potential feedback on climate remain to be assessed at this scale. This is however not the case of the CNRM-CM6-1 climate model (Voldoire et al., 2019), which has been developed at the French National Center for Meteorological Research (CNRM) to be the physical-dynamical core of the CNRM-ESM2-1 Earth system model (Séférian et al., 2019). It is the one model that account
for floodplain processes, and is recognized as offering the most comprehensive representation of groundwater processes in a ESM (Arboleda Obando et al., 2022). Indeed, the simulated land surface accounts for the representation of unconfined aquifer and dynamical floodplains that interact with the root zone soil moisture and the atmosphere through upward capillarity fluxes from groundwater and floodwater evaporation and re-infiltration (Decharme et al., 2019). The goal of the present study is thus to access the potential rule at the global-scale of groundwater and floodplains on present-day climate with the CNRM-CM6-1



climate model. This model and the experimental design are presented in section 2. Section 3 presents the main results while brief discussion and conclusions are given in section 4.

## 2  Method

### 2.1  Model

#### 2.1.1  Basic features

Both the CNRM-CM6-1 climate model and the CNRM-ESM2-1 ESM has been developed at the French National Center for Meteorological Research (CNRM) to contribute to the IPCC Sixth Assessment Report via participation in the 6th phase of the Coupled Model Intercomparison Project (CMIP6; Eyring et al. (2016)). In this study we used an upgraded version of CNRM-CM6-1 in stand-alone mode driven by observed Sea Surface Temperature (SST) described in Colin et al. (2023) . It is based on the ARPEGE-Climat v6.3 atmospheric general circulation model (Roehrig et al., 2020), and the SURFEX v8.0 surface
modelling platform including the Flake lake model (Le Moigne et al., 2016) and the Interaction Soil-Biosphere-Atmosphere (ISBA) land surface model associated to the CNRM version of the Total Runoff Integrating Pathways (CTRIP) river routing model (Decharme et al., 2019). While the CTRIP horizontal resolution is 0.5°, other components use a T127 reduced gaussian grid (about 1.4° in both longitude and latitude). There are 91 vertical levels up to 0.01 hPa in the atmosphere, 14 soil levels down to 12 m and 12 snow levels. A complete description and validation of this model is provided in Voldoire et al. (2019) and
in stand-alone mode in Roehrig et al. (2020).

#### 2.1.2  The land surface

In CNRM-CM6-1, the land surface is represented by the ISBA-CTRIP land surface system described in details in Decharme et al. (2019), and only summarized here. ISBA explicitly solves the one-dimensional Fourier and Darcy laws throughout the soil, accounting for the hydraulic and thermal properties of soil organic carbon. The use of a multilayer snow model of
intermediate complexity allows separate water and energy budgets to be simulated for the soil and the snowpack. In the standard CNRM-CM6-1 model the Leaf Area Index (LAI) was prescribed from the ECOCLIMAP database. In our upgraded version, plant transpiration is controlled by the stomatal conductance of leaves, which depends on carbon cycling in vegetation, i.e. the LAI is prognostic and interacts with climate conditions as in the CNRM-ESM2-1 ESM (Delire et al., 2020; Séférian et al., 2019).

As regards the surface hydrology, a two-way coupling between ISBA and CTRIP is set up to account for : 1) a dynamic river flooding scheme (Decharme et al., 2012) in which floodplains interact with the soil and the atmosphere through free-water evaporation, infiltration and precipitation interception ; and 2) a groundwater scheme accounting for the 218 world's largest unconfined aquifer basins (Vergnes and Decharme, 2012; Vergnes et al., 2014) that combines two-dimensional diffusive groundwater flow between grid cells with vertical upward capillarity fluxes into the superficial unsaturated soil in a fraction
of each grid cell defined by the actual distribution of the sub-grid topography (e.g. Fig.1). The second difference compared





to the standard version of the CNRM-CM6-1 climate model appears in the coupling between the aquifer water table and the superficial soil. While the standard coupling imposes the water table depth to be lower or equal to the hydrological soil depth in ISBA to compute upward capillarity fluxes (Vergnes et al., 2014; Decharme et al., 2019), in our upgraded version the Richards equation is modified to allow the water table to penetrate into the superficial soil (Colin et al., 2023).

## 2.2 Experiments

We performed four experiments consisting in four ensembles of five simulations over the 1980-2014 period driven by observed monthly SST and Sea Ice Cover (SIC). The four experiments correspond to four configurations, noted as followed :

– **ALL** : the groundwater and floodplains schemes are both activated

– **CTL** : the groundwater and floodplains schemes are both deactivated

– **GW** : only the groundwater scheme is activated

– **FLD** : only the floodplains scheme is activated

For each experiment, a transient simulation was run over the 1850-1970 historical period, using restart files from a pre-industrial stabilized simulation run with the same configuration. The 5 members of each experiment were run over the 1970-2014 period, branching from the transient simulation – the ensembles are build using different restart files. The first ten years are considered as spin-up. Results are analyzed over the 1980-2014 period. The various forcings used in CNRM-CM6-1 follows the CMIP6 recommendations Eyring et al. (2016). SST and SIC are given by the Program for Climate Model Diagnosis and Intercomparison (PCMDI) AMIP Version 1.1.3 data set (Durack and Taylor, 2017). Greenhouse gases concentrations are prescribed from Meinshausen et al. (2017) using yearly global averages. Annual mean total solar irradiance forcing are given by Matthes et al. (2017). Many details on other forcings (i.e. tropospheric and stratospheric aerosols) are done in Roehrig et al. (2020).

## 2.3 Statistical significance computations

To assess field differences significance, We use the False Detection Rate (FDR) test from Wilks (2016), with a 95% confidence level. This test, based on local t-tests, allows to reduce the false detection rate (i.e the detection of a signal which is actually not significant) in the case of auto-correlated fields, such as those analyzed in climate science. Further details on this test are given in Wilks (2016) and Colin et al. (2023).

## 3 Results

### 3.1 Groundwater levels and floodplains

In our model, groundwater processes are simulated over the 218 world's largest unconfined aquifer basins, giving access to the water table depth, $z_{wtd}$, over $43\%$ of the total land surface (Fig.1a). In the ALL experiment, $z_{wtd}$ is shallower than $100m$ over



145    40% of the surface covered by the aquifers we represent, while more than a third of this surface exhibits $z_{wtd}$ between $1$ and $10m$, which is coherent with the literature (Fan et al., 2013). On global average, $z_{wtd}$ reaches $19.4m$ depth that is coherent with our estimates using offline simulations in a previous work (Decharme et al., 2019). Similar results are found in the GW experiment (not shown).

### a) ALL Simulated Annual Mean $z_{wtd}$

### b) ALL Simulated Annual Mean $f_{wtd}$

**Figure 1.** Groundwater behaviours simulated by CNRM-CM6-1 (ALL experiment) during the 1980 to 2014 period at 0.5° resolution : **(a)** Annual mean water table depth, $z_{wtd}$ ; and **(b)** Annual mean subgrid fraction of the grid cell allowing vertical capillary rise, $f_{wtd}$, over the same period and at the same resolution. The global mean is given for each panel.





Subgrid fractions, $f_{wtd}$, of each grid cell that allow $z_{wtd}$ to rise into the superficial soil (Fig.1b) represent globally 12%
150   of the area covered by the world's largest unconfined aquifer basins. In some regions known as the flattest in the world (the
Niger delta, the Pantanal, the plains of the Ob basin, along the Amazon, the Ganges valley, the Netherlands, etc.), $f_{wtd}$ can
be significantly higher. Groundwater capillary flux to the unsaturated superficial soil only occurs in lowland near topographic
depressions, rivers or others water bodies (c.f. Fan (2015)). In our groundwater scheme, these lowlands fraction are computed
in each grid cell using the actual distribution of the subgrid topography given by the Global Multi-resolution Terrain Elevation
155   Data 2010 (GMTED2010; Danielson and Gesch (2011)) at a $7.5 - arc - second$ ( $250m$) spatial resolution (see Vergnes et al.
(2014), Decharme et al. (2019) and Colin et al. (2023) for more details).

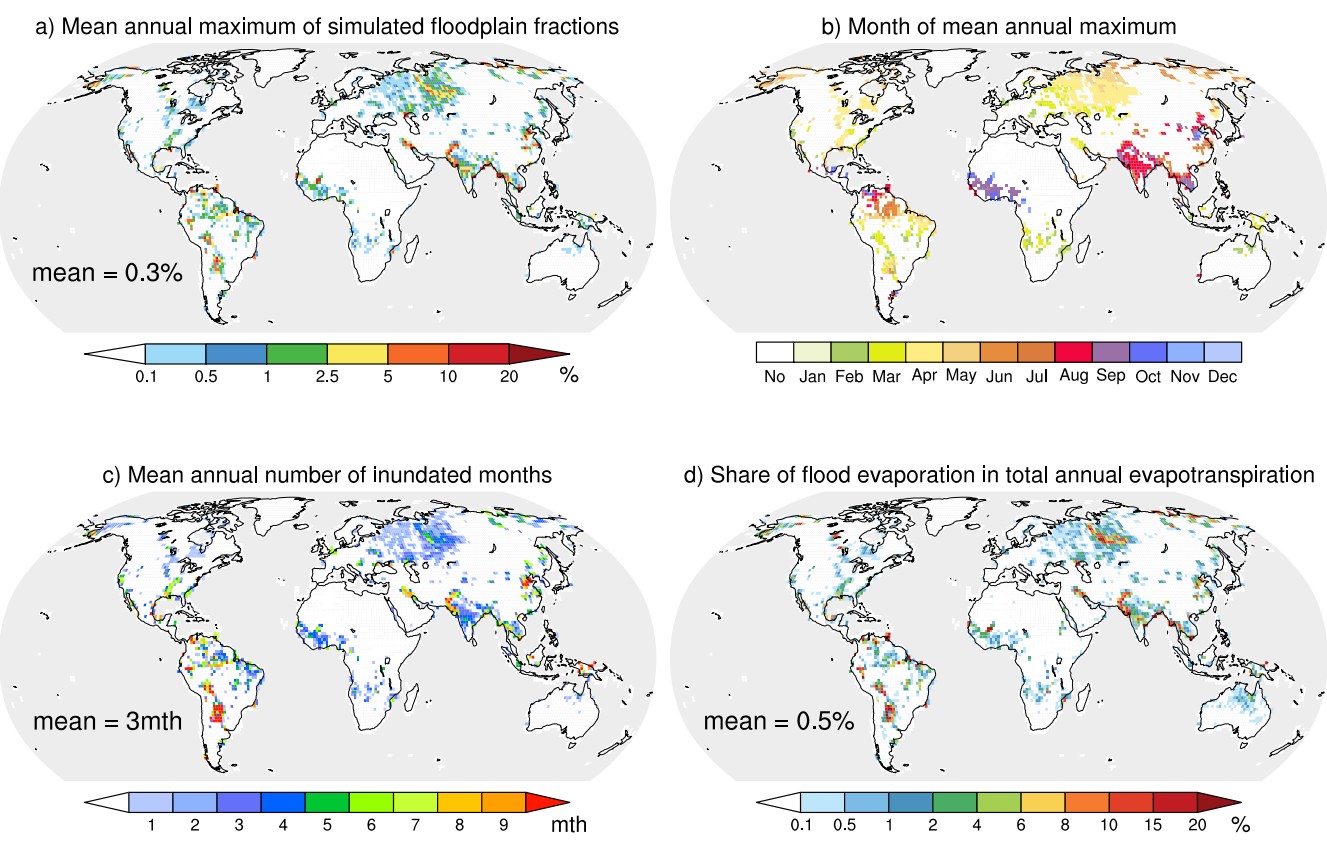

**Figure 2.** Floodplains behaviours simulated by CNRM-CM6-1 (ALL experiment) over the 1980 to 2014 period : **(a)** Maximum fraction of
the mean annual cycle ; **(b)** Month where this annual maximum occurs ; **(c)** Number of month with a non zero inundations fraction ; and **(d)**
Share of floodwater direct evaporation in total annual evapotranspiration.

The seasonal river overflow floodplains simulated by the ALL experiment represents a small fraction of the total land
surface (Fig.2a). Logically, these floodplains are located in the same lowland regions described previously. These inundations





are generally maximum in spring and summer over mid and high latitudes of the northern hemisphere (Fig.2b), that is after
the snowmelt period. It can last between 1 and 3 months and even more along the main rivers (Fig.2c). Over south America
and monsoon regions (west Africa and south Asia), floodplains are maximum after the rainy season and can also last between
1 and 3 months, but can be larger along the main rivers or over Mesopotamia, the Pantanal, the Indus valley, and north China.
Note that this global view of floodplains behavior is coherent with satellite estimates (Prigent et al., 2007) and with previous
offline studies (Decharme et al., 2008; Yamazaki et al., 2011; Decharme et al., 2012). Finally, the share of floodwater direct
evaporation in total evapotranspiration is globally negligible, but can be regionally significant (Fig.2d).

## 3.2 Land surface hydrological impacts

Time variations of the Terrestrial Water Storage (TWS) are very relevant to access impacts of groundwater and floodplains
processes on the simulated land surface hydrology at the global-scale (Vergnes and Decharme, 2012; Decharme et al., 2019).
Indeed, TWS is the sum of all hydrological reservoir presents at the earth surface, i.e. the water on the vegetation canopy, the
snowpack, the inland water bodies, the soil moisture and the groundwater. It is therefore the sum of all reservoirs simulated at
the land surface by a climate model.

Fig.3 presents seasonal spatial patterns estimated by GRACE (see Data availability) and simulated by the CTL and the
ALL experiments. While the simulated patterns appear in good agreement with such estimates, ALL exhibits generally larger
skill scores than CTL (except the bias in MAM) underlying the benefit to simulate groundwater and floodplains processes in
a climate model to study land surface hydrology. This fact is confirmed on the seasonal cycles averaged over Siberia, North
America, Amazonia and South-Asia where the ALL simulated cycles are closer than CTL to GRACE estimates. The GW and
FLD experiments shown on the plots bring some interesting insights about the hydrological mechanisms involved. Over the
tropics and sub-tropics, the improvement from CTL to ALL are mainly related to groundwater processes (GW seasonal cycles
and square correlation closer than FLD to ALL). By storing the water during the rainy season to sustain river base-flow and
surface soil moisture later during the dry season, groundwater processes increase the memory of the system and thus shift
simulated seasonal TWS toward estimates. Over Siberia, and more generally over the north of the northern latitudes, improve-
ments are generally attributable to the representation of river flood. During the snowmelt period, the river water overflows,
and the resulting surplus is stored in large floodplains before it return to the river later in the season. This buffer effect lead
to a positive impact on the simulated seasonal TWS. Over northern mid-latitudes regions, impact of flood and groundwater is
equally distributed.

River discharges are also very relevant to access the simulated water budget over large areas whether using offline land
surface models forced by atmospheric observations or coupled climate models. Beyond the direct evaluation of land surface
hydrological processes, the comparison of simulated river discharges to observations allow to indirectly evaluate the simulated
precipitation over large basins. The climatological seasonal cycles of daily observed and simulated discharges near the mouths
of the world's largest river basins are shown in Fig.4 with the simulated *vs.* observed mean discharge ratio ($r_d = Q_{sim}/Q_{obs}$)
for the CTL and ALL experiments. Three sub-arctic basins (Mackenzie, Ob and Lena), three temperate basins (Mississippi,



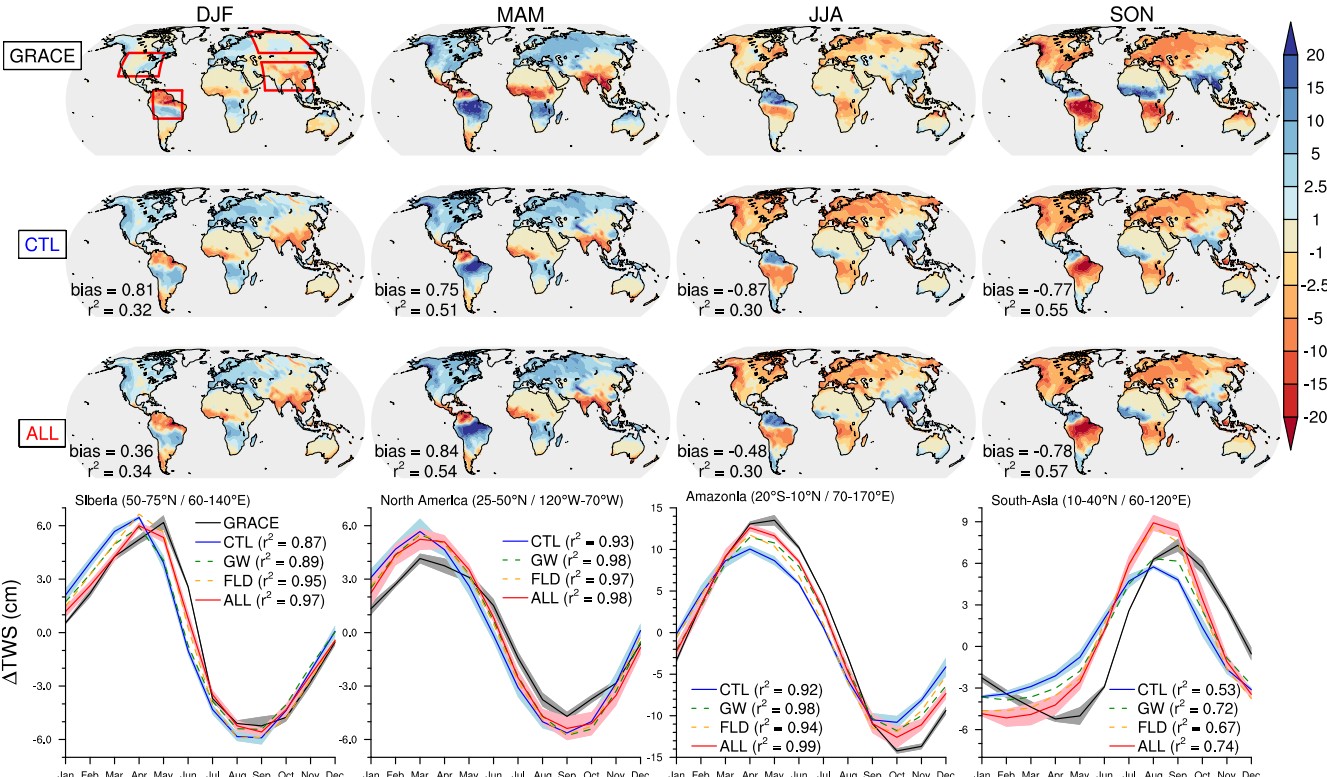

**Figure 3.** Estimated *vs.* simulated seasonal TWS variations (in cm) over the 2002 to 2014 period. Top panels show seasonal GRACE estimates as well as both the CTL and the ALL simulated ensemble means. Global biases and pattern squared correlation ($r^2$) are given for each panel. Bottom plots shows mean seasonal cycles averaged over four regions defined in red in the top left panel. Grace estimates are shown in black where shading correspond to the minimum and maximum values of the ensemble of three GRACE solutions (JPL, GFZ, and CSR). Ensemble means of CTL in blue and ALL in red are given where shading shows $\pm 1.64$ times the intermember standard deviation, while ensemble means of GW (green) and FLD (orange) are shown in dotted line. Square correlation ($r^2$) between estimated and simulated mean seasonal cycles are given for each experiments.

Seine and Danube), three tropical or subtropical basins (Amazon, Congo, Parana), and three monsoon basins (Niger, Ganges, and Mekong) are represented. The simulated *vs.* observed mean precipitation ratio ($r_p = P_{sim}/P_{obs}$) over each basin are also given for the CTL and ALL experiments. Details about river discharge observations are given in Data availability section.

Over sub-arctic basins ALL improve the simulated discharges compare to CTL especially by reducing the lag and the intensity of the summertime peak of discharge. The simulated discharges remain however overestimated ($r_d > 1$) in relation with the general overestimation of the simulated precipitation ($r_p > 1$) over these basins. Whatever, the impact of groundwater and especially floodplains processes are very positive over these regions. Additional GW and FLD experiments confirm that such improvement is mostly due to the floodplain processes. Mechanisms are relatively simple. The floodplain reservoir induces

a buffer effect on river discharge by storing a large part of the springtime snowmelt runoff, thereby limiting the river streamflow



**Figure 4.** Comparison between the mean annual cycle (mm.day$^{-1}$) of simulated and in situ measured daily river discharges near the outlet of the major basins of the world during the 1980–2010 period. Observations are in black, CTL in blue and ALL in red where shading shows $\pm 1.64$ times the intermember standard deviation. The annual simulated discharge ratio to observation ($r_d$) as well as the annual ratio of simulated to observed precipitation rates ($r_p$) are also shown for each basin and model. The observed precipitation rate used to calculate this ratio is the average of the same three products as in Figure 10a. GW and FLD additional experiments are also shown in green and orange, respectively, on each panel.



velocity and thus the lag of the summertime peak of discharge. The reduction of this peak is explains by the direct evaporation of the water stored in the floodplains. This fact is especially noticeable over the Ob basin because the high occurrence of floodplains (Figure 2).

For temperate basins, the mean annual cycles of the observed river discharges are also better reproduced by ALL than by
CTL. The main reason of these improvements is highlighted by the GW additional experiment (green shaded curves in Fig.4). Aquifers store water during the rainy/snowmelt season, when the evapotranspiration is low, and thus contribute to delaying intense river discharges from the spring rainy/snowmelt season to the summer and/or autumn dry seasons (Vergnes et al., 2012; Vergnes and Decharme, 2012; Decharme et al., 2019). Over Europe, the river discharges are however drastically overestimated (cf. Seine and Danube), especially during winter and spring. This weakness is linked to the general overestimation of the
simulated precipitation in these regions over this period (Supplement Figure S1). Conversely, summer precipitation are strongly underestimated balancing the winter to spring overestimation and thus explaining that $r_p$ is close to one over the Danube basin.

River discharge improvements are less obvious over tropical, sub-tropical and monsoon regions, due to a general underestimation of the simulated precipitation (Figure 10a). In general, ALL simulated an annual peak of discharge later in the season than CTL. The resulting smoother mean annual cycles simulated bay ALL seem to be more in phase (and thus in better accor-
dance) with the observations than CTL. Over the Parana, and in lesser extend over the Amazon, the Mekong and the Gange, large floodplains (Figure 2) contribute to plane the annual peak of discharge due to the strong flooded water evaporation (Figure 2d) while aquifers sustain base flow during the dry season as over temperate basins. Over the Congo, aquifer processes are dominant partly because the strong underestimation of the simulated precipitation ($r_p \ll 1$) does not allow to some floodplains to dawn. The case of the Niger is more special. While the ALL simulation appears in better agreement with observations, this
agreement is not very convincing. As discussed in Decharme et al. (2019), the Niger basin has a very complex hydrodynamic structure with : (1) many endorheic sub-basins in the north that do not contribute to the river flow due to a weakly connected drainage network and aridity ; (2) the presence of deep aquifers that can be uncoupled from the river network ; and (3) a large inner delta that favors an intensive evaporation loss and an important aquifer recharge leading to approximately 60% of the inflow lost in the delta. About this last point, even if our model seems to be able to simulate a large inundation in the inner
delta (Figure 2a and 2d), this inner delta is not represented with enough detail to allow a realistic simulation of the Niger basin.

In conclusion, both the $TWS$ variations and river discharges are generally improved by the addition of groundwater and floodplains processes in our climate model. Land-atmosphere coupled models performances to reproduce the observed continental water masses and fluxes are closely related to the quality of the simulated precipitation. In the following subsection, we will see that groundwater and floodplains have a rather limited impact on the model's precipitation biases. So, as already
highlighted in Decharme et al. (2019) using an offline setting of our land surface model, the improvement of the land hydrology with floodplains and groundwater are due to their capacity to increase the hydrological memory of the model, while increasing the total evapotranspiration.



## 3.3 Climate impacts

We now focus on the impact of groundwater and floodplains on the simulated present-day mean climate (i.e atmospheric
variables annually and seasonally averaged over the 1985-2014 period). As showed in the previous subsection, groundwater
and floodplains act as continental reservoirs retaining water which would otherwise flow to the ocean, thus increasing $TWS$.
The bulk of the increase of $TWS$ with groundwater is located well below the root zone and therefore can not be evaporated into
the atmosphere. But when the water table is shallow enough, the presence of groundwater lead to an increase of soil moisture
in the root zone, through the combined effect of capillary rise and a reduced drainage efficiency. Part of this additional soil
moisture can be transported in the atmosphere through surface evaporation and plant transpiration. Floodplains constitutes in
themselves a reservoir of "evaporable" water, but the infiltration of water underneath floodplains also causes an increase of the
root zone water content.

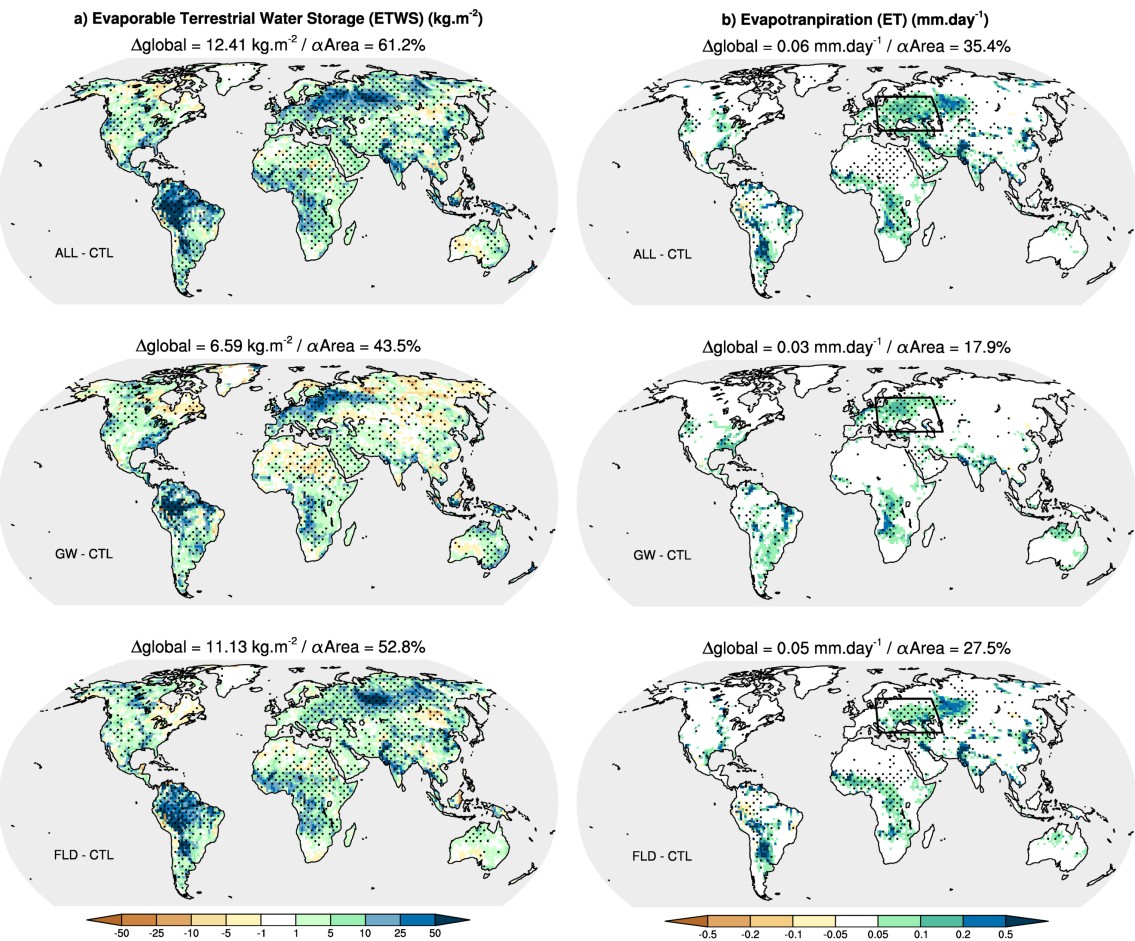

**Figure 5.** Impact of groundwater and flooplains on : (a) Evaporable Terrestrial Water ($ETWS$) ($kg.m^{-2}$), computed as the sum of the
floodplain water storage and the root zone water content, and (b) EvapoTranspiration ($ET$) ($mm.day^{-1}$)



Figure 5 shows the mean annual increase water stored in floodplains and in the root zone (Figure 5.a) along with the increase of evapotranspiration (Figure 5.b) in FLD, GW and ALL experiments, compared to the control experiment (CTL). The amount of evaporable water increases wherever floodplains are present and/or the water table is shallow. Results show that the presence of floodplains has a larger effect on evaporable water than that of groundwater (with a mean increase of 11.13 kg.m$^{-2}$ in FLD and 6,59 kg.m$^{-2}$ in GW). However, most of this increase in FLD and ALL is not due to the additional floodplain reservoir but to the increase of soil moisture in the root zone. In FLD (respectively ALL), water stored in floodplains accounts for 60% (respectively 26%) of the global increase in evaporable water, with an average spatial contribution of 30.7% (respectively 20%) (Figure 6.a). Water evaporated directly from floodplains represents an even smaller fraction of the increase of evapotranspiration (Figure 6.b), with a mean contribution of 17.4% in FLD (10.3% in ALL). Interestingly, the regions where this contribution is highest are generally not those where the increase in evapotranspiration is the largest (Figures 5.a and 6.b). In other words, the increase in evapotranspiration in the presence of floodplains and groundwater is mostly due to a larger water content in the root zone.

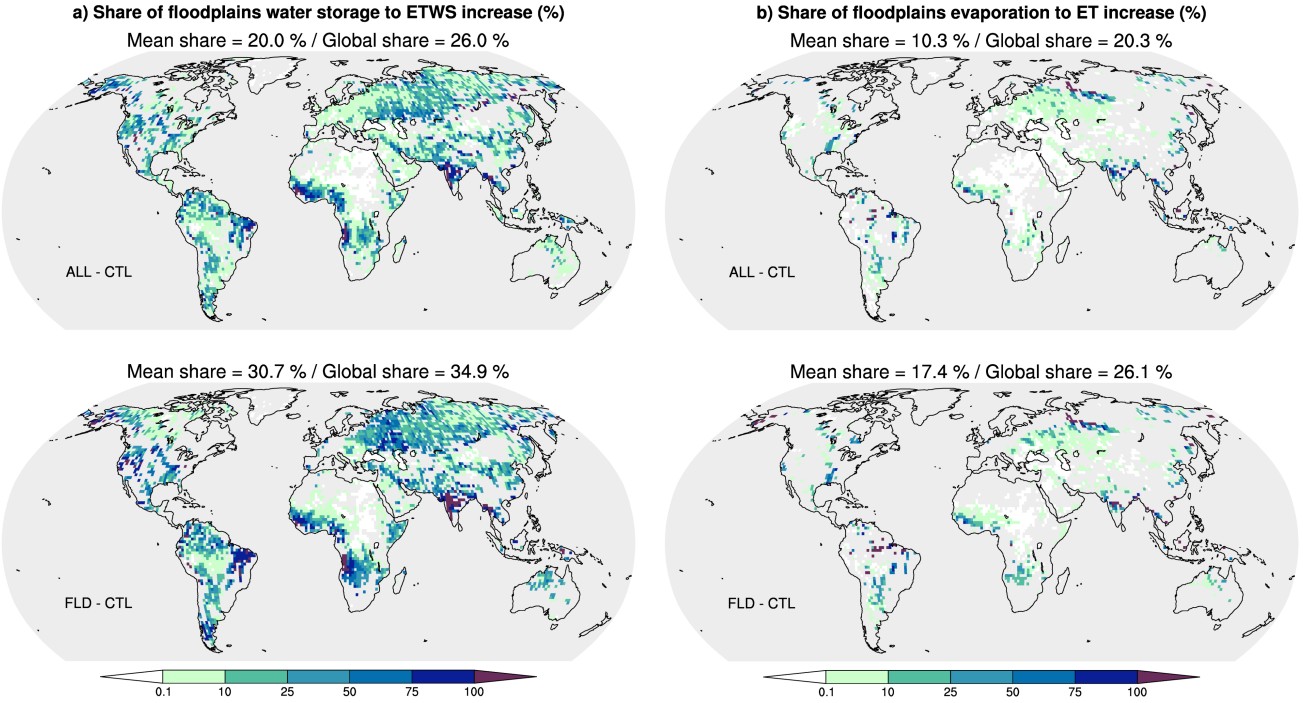

**Figure 6.** Contribution of floodplains to $ETWS$ and $ET$. (a) : Share of floodplains water storage to the Evaporable Terrestrial Water Storage ($ETWS$) shown on Figure 5.a. (b) : Share of floodplains direct evaporation to Evapotranspiration ($ET$) shown on Figure 5.b. The mean share corresponds to the spatial average of grid-points shares. The global shares are computed as the contributions of the spatial sum of the floodplain variables (water content and evaporation) to the spatial sum of the reference variable ($ETWS$ and $ET$)

.



As illustrated in Figure 5.b, the annual increase in root zone water content does not necessarily result in an increase in annual evapotranspiration. It is only when and where the evapotranspiration is limited by soil moisture, rather than by energy, that a larger amount of soil moisture (root zone water content) generates an increase in evapotranspiration Seneviratne et al. (2010). Groundwater can serve as an additional source of soil moisture in regions where evapotranspiration is limited by soil moisture, provided that the region exhibits a marked seasonality of precipitation. In such regions (generally characterised by

high precipitation during the wet season), groundwater recharge can sustain a shallow water table during the dry season. This process is also applicable in areas of complex topography where groundwater converges in valleys Fan (2015); Colin et al. (2023). Similarly, the increase in soil moisture resulting from the infiltration of floodplain water can only lead to an increase in evapotranspiration in regions where the river discharge presents a pronounced seasonal cycle, linked with the precipitation seasonal cycle and/or the existence of a thawing season in the river catchment area.

To further explain the effects of a groundwater and/or floodplains on the atmosphere, we focus on a region in eastern Europe where both groundwater and floodplain impact evapotranspiration (see box drawn on Figure 5.b). Figure 7 shows the seasonal cycles of the differences in liquid soil water content, evapotranspiration components and air relative humidity, spatially averaged over this region, for each simulation with groundwater and/or floodplains (ALL, GW, FLD) compared to the control simulation (CTL) without groundwater nor floodplains . The water content and relative air humidity differences

are plotted along the vertical axis, respectively representing soil depth and altitude). During the recharge season (November to April), the additional soil moisture in the GW simulation is located in the deeper layers of the soil column, which are closer to the water table depth or even beneath it. The upper layers are then either mostly frozen or already close to the field capacity in CTL. As the evaporative season progresses, the soil becomes drier, especially in the upper layers. Aquifers can then provide water to the unsaturated soil layers through capillary rise. Where the water table is shallow, the presence of groundwater also

reduces drainage efficiency, resulting in a larger amount of soil moisture above the water table. This increase of soil moisture in GW leads to an enhancement of evapotranspiration, mostly through transpiration. With more soil moisture in the root zone, plants can transpire more water. Their growth is also enhanced (not shown), resulting in a further increase of the transpiration flux averaged over the region. The increase of leaf area also leads to a larger interception of precipitation by the vegetation, thus increasing the canopy evaporation. The soil moisture added with floodplains (FLD versus CTL) is expectedly larger in

upper layers, as the additional water comes from the inundated surfaces above. It is visible from April to September in this region, where floodplains are present from December to July (see supplementary Figure S2). The floodplain evaporation is maximum in April, when the inundated surface is the largest (supplementary Figure S2). As floodplains dry out, they give way to saturated surfaces, resulting in an increase of soil evaporation in FLD. As the warm season progresses, the vegetation grows and transpiration increases. In summer, the additional transpiration accounts for most of the increase in evapotranspiration in

FLD, as it does in GW. The overall enhancement of evapotranspiration in GW and FLD results in a moistening of the lower troposhpere, with an increase of relative humidity which can be seen from the surface up to approximately 2000 meters. It is strong enough to foster a significant increase of summer precipitation (see supplementary Figure S3), which shows on annual precipitation (Figure 9 for FLD. In return, the enhanced rainfall heightens the increase of soil moisture and evapotranspiration.



**Figure 7.** Mean seasonal cycle differences of relative humidity vertical profiles ($percentage point$) (top), surface water fluxes ($mm.day^{-1}$) (middle), and soil liquid water content vertical profile ($10^{-2}.m^3.m^{-3}$) (bottom), spatially averaged over the "Eastern Europe box" ([15-60°E ; 40-60°N]). Left panel : ALL - CTL ; Middle panel : GW - CTL ; right panel : FLD - CTL. The surface water fluxes considered are the total evapotranspiration (ET) and its components : transpiration (Tran), baresoil evaporation (E soil), evaporation of water intercepted by the canopy (E canopy) and floodplains evaporation (E flood). For each variable, statistically non-significant grid points differences are set to zero for the computation of the spatial average.





Over this particular region, the presence of groundwater has a stronger effect on soil moisture than that of floodplains. However, their respective impacts on evapotranspiration and air humidity have the same magnitude. This can be explained by the fact floodplains affect more the southern part of this region (Figure 5.b), which is drier and thus more sensitive to the increase of soil moisture. The effects of groundwater and floodplains do not linearly add up in ALL, where both groundwater and flooplains are present. On average over the region of Figure 7, the water table is a little shallower in the presence of floodplains (see supplementary Figure S2), as they increase the amount of water infiltrated into the soil. On the other hand, the floodplains surface is smaller in the presence of groundwater (see supplementary Figure S2), as river-groundwater exchanges tend to dampen the river discharge seasonal cycle (see Figure 4), thus reducing the maximum river height and the subsequent inundation. All in all, the combined effect of floodplains and groundwater on the hydrological variables depicted in Figures 7 is less pronounced than the sum of the two individual effects.

Similarly to Figure 7, Figure 8 shows the seasonal cycles of the differences in soil temperature vertical profiles, surface heat fluxes (latent, sensible and downward shortwave) and air temperature vertical profiles of ALL, FLD and GW compared to CTL, spatially averaged over the same box as in 7. As could be expected, the soil temperature is cooler in the simulations with floodplains and/or groundwater (up to -0.5°C). The heat capacity of water being greater than that of soil particles, a wetter soil is cooler (Al-Kayssi et al., 1990), hence the lesser soil temperature with the larger soil water content in ALL, FLD and GW (see Figure 7 during the summer season. This reduced soil temperature results in a decrease of the surface sensible heat flux, which is positive in this region during summer, that is, transporting heat from the surface to the atmosphere. This lessened heat transport in the presence of groundwater and/or floodplains contributes to a cooling of the atmosphere. However, the surface latent heat flux increase is stronger than the sensible heat flux decrease is. This means that the increase of evapotranspiration plays a bigger role in the atmosphere cooling – through a larger heat uptake – than the decrease of sensible heat flux does – through a lessened heat transport from the surface. With the increase of humidity in the atmosphere (see Figure 7) there is also a reduction of the downward surface solar heat flux, which also contributes to the cooling of the low atmosphere. But it is much smaller than the increase of latent heat flux. This shows that most of the decrease in the atmosphere temperature in GW, FLD and ALL is due to the evapotranspiration increase in these simulations. And given that this cooling occurs only in June and July, when the direct evaporation of floodplain water in FLD and ALL is negligible compared to the increase of baresoil evaporation and transpiration (Figure 7), we can conclude that it is mainly driven by the evapotranspiration of the added soil moisture in the presence of groundwater and flooplains. This cooling of the atmosphere can be seen up to an altitude of approximately 2000 meters.

Shifting back to a global perspective, we now consider the effect of groundwater and floodplains on the mean annual precipitation and 2-meter air temperature worldwide (Figure 9). Results indicate that floodplains have a larger impact than groundwater on both these variables, with a much greater percentage of the land surface affected by significant changes and larger mean differences. Focusing on precipitation changes, we find the presence of floodplains (FLD and ALL) leads to an increase of the mean annual rainfall in almost all of the regions of enhanced surface evapotranspiration (Figure 5). This means that in most cases, the increase of evapotranspiration over floodplain regions has enough effect on air humidity to result in larger amounts





**Figure 8.** Mean seasonal cycle differences of air temperature profiles ($K$) (top), surface heat fluxes ($W.m^{-2}$) (middle), and soil temperature vertical profile ($K$) (bottom), spatially averaged over the "Eastern Europe box" ([15-60°E ; 40-60°N]). Left panel : ALL - CTL ; Middle panel : GW - CTL ; right panel : FLD - CTL. The surface heat fluxes considered the turbulent latent (LH) and sensible (SH) heat fluxes, and the downward shortwave radiation (RSdown). The two latter (SH and RSdown) are multiplied by -1. The spatial averages are computed as in Figure 7.





of precipitation. The situation is a little different when considering the addition of groundwater alone (GW). The presence of groundwater does not affect the annual precipitation anywhere, but it does lead to an increase of the mean summer (June to September) precipitation over the Eastern Europe region (Figure S3). Elsewhere, the increase of evapotranspiration induced by groundwater is not large enough and/or does not occur in the right season to affect precipitation. Conversely, the regions of increased precipitation are all affected by an increase of evapotranspiration, except for the central and northern parts of the amazonian basin, where the annual rainfall rate is larger in the presence of floodplains (ALL and FLD) without it being associated with an increase of the annual evapotranspiration - in fact, we find a decrease of evapotranspiration. Further analysis suggests that over this region, humidity is transported from the floodplains of Venezuela, the mouth of the Amazon river and/or northern Bolivia (not shown). The subsequent increase of annual precipitation leads to a larger mean cloud cover and a decreased downward solar radiation (see figure S4.c). And since over this humid region, evapotranspiration is limited by energy rather than by soil moisture, we find a decrease of evapotranspiration associated with the increase of precipitation. Everywhere else, the effects of groundwater and/or floodplains on precipitation remain circumscribed to the areas of evapotranspiration increase, meaning that the additional continental water induced by the presence of floodplains is transported only vertically in the atmosphere.

If we now consider the impact of groundwater and floodplains on 2-meter temperatures (Figure 9), we find a cooling of the near surface atmosphere in most of the regions where the surface latent heat flux (i.e. evapotranspiration) is affected along with the surface sensible heat flux (see Figure S4). Comparing the amplitudes of latent and sensible heat flux differences, we find that the increase of the former is larger than the decrease of the latter by a factor 1.25 to 2, everywhere except over a few grid cell in eastern Sahara, where the differences are very small for both fluxes and the precipitation is barely affected. The downward surface solar radiation is also reduced with the presence of floodplains (FLD and ALL) in some of the regions of increased air humidity in the lower atmosphere (Figure S4). However, the amplitude of this decrease of solar radiation remains limited (1 to 2 W.m-2) and it mostly affects regions where the atmosphere cooling is not significant. In the few areas where the decrease of solar radiation is combined with a reduction of the 2-meter temperature, we find that it is 1.5 to 3 times smaller than the increase of surface latent heat flux. Therefore, the decrease of 2-meter temperature simulated in the presence floodplains and/or groundwater is mostly due to evaporative cooling. This effect is stronger on warm temperatures, as the atmosphere evaporative demand increases with temperature. And indeed, we find a larger greater cooling of mean summer temperatures (compared to mean annual temperatures), as well as on mean daily maximum temperatures (compared to mean minimum temperatures), as shown on Figure S5 for the boreal summer mean values.

Over some regions, the cooling of the lower atmosphere in the presence of floodplains (FLD and ALL) is associated with a slight increase of the sea level pressure mostly in the northern hemisphere during the boreal summer (see Figure S6). However, this effect on pressure quickly fades with the altitude, as there is no significant impact of groundwater or floodplains on the geopotential heights above the 925 Pa level (not shown). We also found no significant impacts on wind components mean values, regardless of the altitude (not shown). Thus, groundwater and floodplains have no significant impact on the atmosphere dynamics in our simulations, at least from a climatic perspective (that is, on annual and seasonal averaged over a fairly long





**Figure 9.** Impact of groundwater and floodplains on simulated annual mean **(a)** precipitation and **(b)** 2m air temperature over the 1980–2014 period. Ensemble mean differences between CTL and ALL (first row), CTL and GW (middle row), and CTL and FLD (last row) are shown with theirs global values (Δglobal). The stippling indicates regions with statistically significant difference between CTL and ALL at a 95% level of confidence using the FDR test. The % of the continental area (excluding Antarctica) which is statistically significant (αArea) is also given for each panels





period). And as mentioned before, their effect on the atmosphere temperature, humidity and precipitation remain "local" almost everywhere, in the sense they occur above land surfaces affected by an increase of evapotranspiration. In other words, there is no significant advection of the temperature and humidity changes, except in central and southern Amazonia, where the

additionnal air humidity is transported from floodplains located in regions nearby. To sum up, representing groundwater and floodplains in our model leads to an increase in precipitation and a cooling of warm temperatures in a number of regions worldwide, with a larger impact of floodplains. Whether or not these impacts improve the model's biases is addressed in the following paragraphs. This last piece of our analysis does not include a thorough discussion on the biases sources, as this has already been done in Roehrig et al. (2020).

The model's annual precipitation biases are shown in Figure 10a for the CTL and ALL experiments (details about precipitation observations are given in Data availability section. As already pointed out (section 3.2), whatever the experiment, a drastic underestimations of these precipitation is found over the tropics. Elsewhere, the amplitude of the annual bias is much more acceptable. However, these biases can vary greatly depending on the season (Supplement Figure S1). For example, over central Eurasia or the US great plains, winter precipitation is generally overestimated while summer precipitation is underestimated.

Comparing the CTL and ALL biases in regions where the differences between CTL and ALL are statistically significant (Figure 10a), we find that groundwater and floodplains reduce the underestimation of precipitation in Africa, South America and central Eurasia. This amounts to a mean global improvement of precipitation and a larger land surface area impacted by an improvement than by a worsening of the bias. In central Eurasia, the bias reduction is due to the increase of summer precipitation (June to September), which can be attributed mostly to groundwater in the western part of the region and to floodplains

in the eastern part (see Supplement Figures S3 and S1). In South America, the increase of precipitation is almost entirely due to floodplains and occurs during the austral summer (December to March) (Supplement Figures S3 and S1). In Africa, the improvement is again due to floodplains and it occurs during the rainy season, which corresponds to the boreal summer above the equator (Supplement Figures S3 and S1) and spans from October to February in southern Africa (not shown).

The annual biases of 2-meter air temperatures are shown in Figure 10b for the CTL and ALL experiments (see Data avail-

ability for details on observations). A warm bias is found over the Pantanal region, central and southern Africa, eastern Siberia, central Eurasia and Australia while Greenland and the Himalayas show a cold bias. In the northern hemisphere, the cold biases occur in winter and the warm ones are seen mainly in summer, except for eastern Siberia, which shows a warm biais in winter (Figure S7). In the Pantanal region and Australia, the warm bias persists throughout the year, with a larger amplitude during the austral summer (December to March). In central Africa, the warm bias is stronger during the austral winter (June to

September) and in southern Africa, the warm bias is only seen during this season (Supplement Figure S7). As is the case for precipitation, the impact of groundwater and floodplains on the 2-meter air temperature biases is mostly positive. It consists in a reduction of the warm biases found in the Pantanal throughout the year and in central Eurasia in summer (Supplement Figure S7). In the Pantanal, the improvement is almost entirely due to floodplains. In central Eurasia, the cooling induced by the presence of groundwater reduces the bias over the western part of the region while floodplains cool the eastern part (Figure





**Figure 10.** Observed *vs.* simulated annual mean **(a)** precipitation and **(b)** 2m air temperature over the 1980–2014 period. First row panels show observations and theirs global mean values. Panels of middle rows show both the CTL and the ALL ensemble mean biases compared to observations with theirs global biases and pattern square correlations ($r^2$). Last row panels shows differences between these CTL and ALL absolute biases over regions with statistically significant difference between CTL and ALL at a 95% level of confidence using the FDR test (cf. section 2.3). A negative value (green) means an improvement (i.e. ALL closer to observations than CTL) and conversely for a positive value (purple). The % of the continental area (excluding Antarctica) where this difference is statistically significant ($\alpha$Area), the global mean of this difference ($\Delta$) over this area, and the % of the continental area with negative (-) and positive (+) values are also given.



9). During the boreal summer, the combined effects of floodplains and groundwater also reduce the warm bias over parts of the US (Supplement Figures S7 and S5).

As previously explained, the decrease of air temperature and increase of precipitation induced by floodplains and groundwater is mostly due to the additional evapotranspiration and air humidity. In the regions where this leads to a reduction of the temperature and precipitation biases, we find that the model's evapotranspiration and air humidity are mostly underestimated
in the CTL experiment and that this dry bias is improved in ALL (Figure 11 and Supplement Figures S8 and S9). So the impact of groundwater and floodplains on the model's biases of temperature and precipitation biases does not result from a compensation of errors. The representation of groundwater and floodplains does really improve our model's realism by adding missing processes.

## 4    Discussion and Conclusion

The inclusion of floodplain and groundwater processes in the CNRM climate model has demonstrated significant improvements in the simulation of land surface hydrology and climate in a number of regions. One of the primary impacts on land hydrology is the enhanced accuracy in terrestrial water storage variations and river discharges. This improvement is attributed to the increased hydrological memory and surface evapotranspiration brought about by the integration of groundwater and floodplain dynamics. As already highlighted in Decharme et al. (2019) using an offline setting of our land surface model, the extended
hydrological memory allows the model to better represent the seasonal variability of water storage and flows, leading to more accurate simulations of river discharge and terrestrial water storage. Despite these positive outcomes, the model reveals several notable discrepancies when compared to observed river discharge data. These deficiencies can be attributed to simulated precipitation biases in certain regions, but they can also be attributed to model weaknesses. For example, the Niger basin is poorly simulated, which is likely due to a lack of representation of its highly complex hydrodynamic structure. In regions of
the world where these issues are prevalent, basin-by-basin work will be essential in the future to advance the representation of hydrological processes in our climate model.

Regarding climate variables, the inclusion of these processes has led to increased surface evapotranspiration and in regions where this flux is primarily driven by soil moisture. This result in a decrease in near-surface air temperatures, particularly warm ones, and an increase in precipitation in those areas, effectively reducing the model's biases on these variables. Im-
portantly, these improvements are not the result of error compensation. They are based on more realistic representation of land-atmosphere interaction as evidenced by the comparison of simulated evapotranspiration and air humidity with observational data, showing that biases in these variables are also reduced with the inclusion of groundwater and floodplains. The processes involved here are consistent with previous studies showing that inland water bodies can influence the atmosphere by humidifying and cooling its lower layers throughout an increase in evapotranspiration (Bonan, 1995; Lofgren et al., 2002;
Krinner, 2003; Balsamo et al., 2012; Le Moigne et al., 2016; Arboleda Obando et al., 2022).





**Figure 11.** Observed *vs.* simulated annual mean **(a)** 2m air relative humidity over the 1980–2014 period and **(b)** land surface evapotranspiration over the 1982–2008 period. First row panels show observations and theirs global mean values. Notations are the same than Figure 10. Relative humidity differences are given in percentage points (pp).





It is, however, important to note that the overall changes induced by floodplains and groundwater on atmospheric climate variables remain relatively small when compared to the existing biases. This suggests that while these processes significantly enhance the model's hydrological components, their impact on climate variables is more modest. Nonetheless, the improvements in land hydrology processes are significant, which in itself justifies the inclusion of these processes in global climate
models. Moreover, incorporating these processes in a coupled model framework allows for the representation of feedback mechanisms between the land surface and the atmosphere. These feedbacks are essential for understanding the complex interactions that drive climate dynamics. By capturing these interactions, the model can provide more comprehensive insights into the hydrological and climatic processes, thereby improving the overall quality of climate simulations.

In addition to the water cycle, incorporating these processes in ESMs is important to study the impact of climate and
hydrology on the carbon cycle and vice versa. Soil moisture is a key driver of natural $CO_2$ emissions, regulating both plant behavior and the microorganisms that decompose organic matter in the soil (Canadell et al., 2021; Friedlingstein et al., 2023). However, the impact of groundwater on soil decomposition and carbon emissions remains largely unexplored to date, even though it acts as lower boundary conditions for soil moisture. Using groundwater schemes in ESMs could help fill this gap. The use of floodplains scheme in ESMs is also important for understanding the interplay between hydrology and methane emissions,
and especially at the inter-annual timescale. Floodplains, with their dynamic water levels and saturated soils, are ecosystems poor in oxygen. These anaerobic conditions favour methane production from the decomposition of soil organic matter (Saunois et al., 2020; Morel et al., 2019). Hence simulating the distribution and variations of such surface water bodies over the global land surface in ESMs is also a key question regarding climate change and projection of greenhouse gas emissions.

Finally, anthropogenic processes such as irrigation and dams that can alter the river flow and increase the continental evap-
otranspiration (Sacks et al., 2009) are not accounted for in our model. Looking ahead, our next step is to incorporate these anthropogenic water fluxes into the model (Druel et al., 2022; Sadki et al., 2023; Decharme et al., 2024). This addition aims to further enhance the representation of land hydrology and its interactions with climate. By doing so, we hope to achieve an even more accurate and comprehensive simulation of the coupled land-atmosphere system, ultimately contributing to better predictions and understanding of future climate scenarios.

*Data availability.* The results of all the models examined here are available at https://zenodo.org/records/13882951 (Decharme and Colin, 2024). Some details about the reference data sets used for model evaluation is also done hereafter : **Terrestrial Water Storage variations -** Changes in continental water masses can be detected from space using the Gravity Recovery And Climate Experiment (GRACE) solutions (Swenson, 2012). Here, we used TWS monthly dynamic estimated at a monthly frequency over the 2002–2014 by three GRACE solutions at 1° resolution : the RL05 GRACE release provided by the Center for SpaceResearch (CSR) at the University of Texas at Austin, the
Jet Propulsion Laboratory (JPL), and theGeoForschungsZentrum (GFZ) at Potsdam. GRACE land are available at http://grace.jpl.nasa.gov, supported by the NASA MEaSUREs Program ; **River discharges -** We evaluate daily simulated river discharge at major river outlets using in situ measurements provided by the Global Runoff Data Centre (GRDC) available at http://www.bafg.de/GRDC/EN/Home/homepage_node.html, along with the United States Geological Survey (USGS) data for the Mississippi basin available at http://waterdata.usgs.gov/



nwis/sw, the French Hydro database for the Seine basin available at https://naiades.eaufrance.fr/, the HYdro-géochimie du Bassin AMa-
zonien (HyBAm) data over the Amazon basin, and the streamflow time series of the Parana river at Rosario from Antico et al. (2018) ;
**Precipitation -** Three product are used at a monthly frequency over the 1980–2014 period to evaluate the simulated precipitation : the
Global Precipitation Climatology Centre (GPCC) product (Schneider et al., 2014) available at https://climatedataguide.ucar.edu/climate-data/
gpcc-global-precipitation-climatology-centre, the Global Precipitation Climatology Project (GPCP) product (Huffman et al., 2009) avail-
able at http://gpcp.umd.edu/, and the Multi-Source Weighted-Ensemble Precipitation (MSWEP) product (Beck et al., 2017) available at
http://www.gloh2o.org/mswep/ ; **2m air temperature -** Three product are used at a monthly frequency over the 1980–2014 period to eval-
uate the simulated 2m air temperature : the Berkeley Earth Surface Temperature (BEST) product (Rohde and Hausfather, 2020) available
at https://berkeleyearth.org/data/, the Climate Research Unit gridded Time Series version 4.06 (CRU-TS4.06) product (Harris et al., 2020)
available at https://crudata.uea.ac.uk/cru/data/hrg/, and the Global Historical Climatology Network version 2 and the Climate Anomaly Moni-
toring System (GHCN-CAMS) product (Fan and van den Dool, 2008) available at https://psl.noaa.gov/data/gridded/data.ghcncams.html ; **2m**
**air relative humidity -** Three product are used at a monthly frequency over the 1980–2014 period to evaluate the simulated 2m air relative hu-
midity : the CRU-TS4.06 product (Harris et al., 2020) available at https://crudata.uea.ac.uk/cru/data/hrg/ of vapor pressure that we combined
with the 2m air temperature product using the Clausius–Clapeyron relationship, the Met Office Hadley centre Integrated Surface Database
Humidity (HadISDH) version 4.4.0.2021f (Willett et al., 2014) available at https://www.metoffice.gov.uk/hadobs/hadisdh/, and the ERA-
Interim reanalysis (Dee et al., 2011) available at https://www.ecmwf.int/en/forecasts/datasets/reanalysis-datasets/era-interim ; **Land surface**
**evapotranspiration -** Global evapotranspiration estimates are given at a monthly frequency over the 1982–2008 period by three independent
products : the Multi-TreeEnsemble (MTE) product derived from satellite data and FLUXNET in situ observations (Jung et al., 2010) available
at https://climatedataguide.ucar.edu/climate-data/fluxnet-mte-multi-tree-ensemble, the TerraClimate land surface evapotranspiration product
(Abatzoglou et al., 2018) reconstructs from observations and reanalysis available at https://www.climatologylab.org/terraclimate.html, and
the Global Land Evaporation Amsterdam Model (GLEAM) version 3.6a product based on satellite and reanalysis data (Martens et al., 2017)
available at https://www.gleam.eu/.

*Code and data availability.*  The CNRM-CM6-1 climate model source code is not freely available but a detailed description can be found at
https://www.umr-cnrm.fr/cmip6/?article11&lang=en.

*Author contributions.*  The article was written by BD and JC. BD and JC developed the study design. BD supervised or developed the
groundwater and flooding schemes, as well as all couplings between hydrology and atmosphere in CNRM-CM6-1. JC ran the experiments.
BD and JC analysed the results.

*Competing interests.*  The contact author has declared that none of the authors has any competing interests.



*Acknowledgements.* This work is supported by the "Centre National de Recherches Météorologiques" (CNRM) of Méteo-France and the "Centre National de la Recherche Scientifique" (CNRS) of the French research ministry. Additional support was provided by the European Union's Horizon 2020 (H2020) research and innovation program under Grant Agreement No. 101003536 (ESM2025-Earth System Models for the Future)




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
