# Peer review of "Influence of Floodplains and Groundwater Dynamics on the Present-Day Climate Simulated by the CNRM Climate Model"

_EGUsphere, 2024_

## Community Comment (CC2)

[Figure]

**Figure 1 -** Schematic view of the CNRM-CM6-1 climate model in "AMIP mode", that is not coupled to the NEMO ocean model. The model couples the ARPEGE-Climat v6.3 general circulation model with the ISBA-CTRIP system to simulate atmospheric, land surface, and hydrological processes. The ISBA land surface model includes a one-layer vegetation interception scheme, $CO_2$-responsive plant transpiration, bare soil evaporation, snow sublimation, a 12-layer snow scheme, and a 14-layer explicit soil scheme for temperature, moisture, and soil ice. Soil moisture is computed within the rooting depth, which varies from 0.2 to 8 meters depending on the vegetation type, while soil temperature is simulated down to 12 meters. Subgrid hydrology accounts for surface runoff, deep drainage, and recharge fluxes, with groundwater interactions modeled over the low land fraction of the grid cell affected by the water table, $f_{wtd}$ (illustrated by the grey footprint around the river). The CTRIP hydrological model simulates river discharge using total runoff from ISBA, dynamically solving river velocity with Manning's formula and incorporating a flooding scheme to compute flood volume and extent. It also includes a two-dimensional groundwater scheme for unconfined aquifers, representing time variations in water table depth, interacting with rivers and fed by the recharge rate simulated by ISBA. The coupling between ISBA and CTRIP is achieved via the OASIS-MCT interface, allowing interactions between floodplains, soil, and atmosphere through evaporation, infiltration, and precipitation interception, as well as upward capillary fluxes into the unsaturated soil over $f_{wtd}$. Variables involved in these coupling processes are shown in black. Further details are available in Decharme et al. (2019).

---

## Author Response (AR1)

**Response to referees' comments for manuscript egusphere-2024-3091**

The authors would like to thank the reviewers for their constructive comments and suggestions. The manuscript has undergone a thorough revision according to the reviewers' comments. Please see below our responses. Note that we slightly changed the title to add "Climate" with CNRM Model. So the new tittle is not so different (difference in red): "Influence of Floodplains and Groundwater Dynamics on the Present-Day Climate simulated by the CNRM Climate Model". Figure 1 has been added as required by the reviewer 2, so all figure number changed.

**Response to the reviewers**
* * *
**Reviewer 1**

**General Comment** — This manuscript describes the results of the climate model experiment to discuss the potential impact of representing groundwater and floodplain. The experiment is well designed, and the simulation data are appropriately analyzed. The finding that the representing floodplains and groundwater improve the hydrological cycle and lower atmosphere status (precipitation and temperature) is interesting. I don't find any critical errors in the manuscript, thus I feel the paper can be accepted after a minor revision.

**Reply**: Thank you very much for your review and useful comments. We changed the manuscript to include your corrections and clarify the point regarding $f_{wtd}$.

**Reviewer Comment 1.1** — Line 13: predominantly positive
This description is ambiguous, and I cannot get what changes were observed. Please include more easy-to-understand description. (e.g. "bias is reduced").
Line 12: The study found:
I think it might be better to explain the finding that "the simulated hydrological cycle was improved by representing floodplain and groundwater (Fig 3 and 4)" also in the abstract.

**Reply**: Regarding your first two comments, we followed your suggestions and changed the sentences in the abstract (lines 10-15).
In the first version of the manuscript, it read : "*This study found that the impact of groundwater and floodplains on precipitation and 2-meter air temperature biases is predominantly positive in comparison to observations, although on a very regional scale. Additionally, the model's ability to simulate terrestrial water storage and river discharges is enhanced by these processes. The improvement in land hydrology with floodplains and groundwater is attributed to their ability to increase the hydrological memory of the model.*"

It now reads : *"The simulated hydrological cycle is improved by representing floodplains and groundwater, thanks to an increased hydrological memory which allows to better capture the seasonal cycle of the terrestrial water storage and river discharge. Additionally, the inclusion of groundwater and floodplains reduces precipitation and 2-meter air temperature biases at the regional scale."*

**Reviewer Comment 1.2** — L148: Figure 1b Is the subgrid fraction $f_{wtd}$ is variable in time or constant in time? From the sentence, it looks like the $f_{wtd}$ is the parameter calculated by the topography. While in the figure, it is labelled as "All simulated annual mean $f_{wtd}$", and I feel this is variable in time. If $f_{wtd}$ is a time-constant parameter, I think it's better to show that in the model description section rather than the result.

**Reply**: Yes, $f_{wtd}$ is variable in time, as it depends on wtd. We clarified this point in the manuscript. In the model's description section, lines 114-115, we changed the sentence " *with vertical upward capillarity fluxes into the superficial unsaturated soil in a fraction of each grid cell defined by the actual distribution of the sub-grid topography* " into " *with vertical upward capillarity fluxes into the superficial unsaturated soil allowed over a fraction of each grid cell which varies with the water table head and its position with respects to the sub-grid topography distribution* ".
Also, line 148, the sentence " *Subgrid fractions, $f_{wtd}$, of each grid cell that allow $z_{wtd}$ to rise into the superficial soil* " was changed into " *The temporal mean values of $f_{wtd}$ the fraction of each grid cell over which $z_{wtd}$ is allowed to rise into the superficial soil* ".

**Reviewer Comment 1.3** — L214: simulate bay ALL "bay" should be "by"

**Reply**: We also corrected the typo line 214 ("bay" instead of "by").
* * *
**Reviewer 2**

**General Comment** — This article is a fit for Earth System Dynamics given its focus on how the inclusion of floodplain and groundwater modules impact Earth System Model (ESM) results, linking improved hydrologic cycle representation to reduced bias in a number of land and atmosphere variables and usefulness in carbon cycle studies. The study is novel in the sense that the global scale interaction of the atmosphere with inundated areas has not been previously studied. Nor has a global study of the impact of realistically rendered groundwater aquifers been carried out with an ESM before.
The results of this comprehensive study are presented with clarity and sufficient detail overall. The experiment description is complete enough to allow for reproducibility. The title is also clear, but could potentially be revisited to include the term "Hydrology" as the impact of the floodplain and groundwater modules of land surface hydrology is a major part of this

study. The supplement provides a useful deeper dive into the results and are referenced in a useful way within the main text.

**Reply**: Thank you for your specific comments which helped us clarify the manuscript by giving more details on the points you commented on. We also thank you very much for having taken the time to make all these corrections on typoes and english mistakes. For the sake of clarity in our answer below, we copied each of your comments in bold characters before answering them.

**Reviewer Comment 2.1 —**

1. The methods used in this study are clearly outlined overall, with a small handful of requests for added clarity to be found within this section of the peer review.

2. Will the authors please add a description of GRACE and how it is used in the methods section? It would also help to know the rationale for selecting GRACE over other comparable products. And if other comparable products exist, it would also be good to learn why an observational ensemble is not being used. To be clear, I think it's okay to use just one observational dataset, it would just be good to know why this choice has been made.

3. Also related to the methods section, will the authors include a sentence to indicate why performing this set of experiments in offline mode is preferred?

4. On line 174, skill scores are referenced. I assume this is related to the bias scores presented in Figure 3, but is it possible to state which test of skill is used and refer to the corresponding figure within the main text to make this more explicit?

5. Could figure 11 be moved to before the discussion and conclusion section?

6. Is it possible to include a model structure diagram to show how CRNM-CM6-1 incorporates the ISBA and CTRIP models, as well as sub-grid processes? This might help other ESM modelling teams envision how floodplain and groundwater processes could be incorporated into their own models.

7. Figure 5b seems to appear before it is mentioned in the main text. Is it possible to move this figure (or the mention of it) so that this is no longer the case?

8. Ask for confirmation of what the model uses as maximum rooting depth

9. Please include a sentence on why the False Detection Rate test was used to test for statistical significance instead of other sign-based tests.

**Reply**:

1. OK

2. We added information on the GRACE data and the reasons why we chose it at the beginning of section 3.2 (after line 171) : "*The Gravity Recovery and Climate Experiment (GRACE) satellite mission, developed by NASA and the German Aerospace Center (DLR), measures temporal variations in the Earth's gravity field. These variations are used to estimate changes in terrestrial water storage (TWS), including surface water, soil moisture, and groundwater. Therefore, we used these GRACE data to evaluate the TWS variations simulated by our climate model (see also Data availability). This dataset consists of an ensemble of three monthly TWS estimates, from 2002 to 2014; they are derived from solutions provided by three independent processing centers: the Jet Propulsion Laboratory (JPL), the German Research Centre for Geosciences (GFZ), and the Center for Space Research (CSR). These data are uniquely suited for evaluating large-scale hydrological processes as they provide a global coverage of direct measurements which can not be obtained through other observational products.*" However, this was not done in the methods section, where we chose not to include a "dataset" subsection, given that all the datasets we used are very well known, accessible and documented in the references we give in the "data availability" section.

3. All the simulations analyzed in the paper were run in "inline" mode, that is with the land surface model coupled with the atmospheric model. Did you mean "inline" instead of "offline" ? Or by "offline", you meant "not coupled to the ocean model"? At any rate, we specified again in the methods section (2.2) that the simulations were run with CNRM-CM6-1, not coupled to the ocean model. The simulations were run inline because the paper deals with the effects of groundwater and floodplains on the atmosphere. They may also have a limited impact on the ocean dynamics near the rivers mouth, but this is beyond the scope of the paper. It is very unlikely that the expected limited impact on the ocean would affect the atmosphere, and running the same experiments in a fully coupled mode (that is, coupled to the ocean model used in CNRM-CM6-1) would require a much larger number of members as the model's internal variability is much greater in this configuration, so we felt it was unecessarily expensive.

4. "Skill scores" in this sentence actually refers to the mean biases and root mean square errors, shown on Figure 3. We clarified it in the manuscript.

5. Agreed. It is usually the journal that takes care of the final editing and positionning of the figures. We will make sure the figures are well positioned in the proofreading file.

6. We did. This new figure and its descriptive caption is attached to this reply (page 14).

7. Agreed. It is usually the journal that takes care of the final editing and positionning of the figures. We will make sure the figures are well positioned in the proofreading file.

8. It depends on vegetation types. We added a paragraph describing this point in the methods section (2.1.2) : "The soil moisture profile is computed within the rooting depth, which varies from 0.2 to 8 meters depending on the vegetation type, as detailed in Figure 2 and Table 1 of Decharme et al. 2019.These rooting depths values are derived from the

ECOCLIMAP database. They reflect vegetation-specific adaptations to climatic and soil conditions. In contrast, the soil temperature profile is computed down to a depth of 12 meters to account for thermal dynamics beyond the rooting zone. To represent land cover and rooting depth heterogeneities, a tile-based approach is employed. This approach allows multiple vegetation types to coexist within a single grid cell. Distinct energy and water budgets are computed for each tile, and their relative fractional coverage within the grid cell is used to determine the grid-box-averaged water and energy budgets."

9. This was done in section 2.3 of the method section, where we state "To assess field differences significance, we use the False Detection Rate (FDR) test from Wilks (2016), with a 95% confidence level. This test, based on local t-tests, allows to reduce the false detection rate (i.e the detection of a signal which is actually not significant) in the case of auto-correlated fields, such as those analyzed in climate science. Further details on this test are given in Wilks(2016) and Colin et al. (2023)."
To address your comment, we added a sentence to this paragraph. The paragraph now reads : "To assess field difference significance, we use the False Detection Rate (FDR) test from Wilks (2016), with a 95% confidence level. It relies on t-tests performed for each grid point. But instead of comparing the p-values to a fixed value (0.05 for a 95% confidence level) as it is classically done, the FDR test consists in computing a threshold which also corresponds to a given confidence level (here 95%) but depends on the series of all P-values. This allows to reduce the false detection rate (i.e the detection of a signal which is actually not significant) in the case of auto-correlated fields, such as those analyzed in climate science. Further details on this test are given in Wilks(2016) and Colin et al. (2023)."

**Reviewer Comment 2.2** — Please consider the following technical corrections.

1. Page 1: Consider capitalizing "S" in "simulated" for consistency with title case used throughout the rest of the title.

2. Page 2, line 20: Delete "continental".

3. Page 2, line 22: Delete "front of" and replace with "the face of".

4. Page 2, line 30: Delete the "s" at the end of "represents".

5. Page 2, line 32: Add an "s" to the end of "term".

6. Page 2, line 33: Delete the "s" at the end of "discharges".

7. Page 2, line 40: Depending on the intended meaning, it may be clearer to say "potential evaporation" rather than just "potential".

8. Page 2, line 47: The word "partitioned" may be more accurate that "distributed" here.

9. Page 3, line 53: Delete "the" after "North America, ".

10. Page 3, line 54: Replace "or" with "and" before "east Africa".

11. Page 3, line 54: Delete "these" before "water bodies".

12. Page 3, line 57: Delete "s" after "impacts".

13. Page 3, lines 57 and 58: Reverse the order of "enough documented" to read "documented enough".

14. Page 3, line 64: Delete "an" before "idealized".

15. Page 3, line 68: Delete the "s" at the end of "theirs".

16. Page 3, line 69: The word "capacity" may be preferred to "capability" in this context.

17. Page 3, line 70: The word "role" may be more appropriate than "rule" in this sentence.

18. Page 3, line 71: Consider including a comma after "rivers" to indicate that a phrase is being added to the main clause in this sentence.

19. Page 3, line 72: Delete the "s" at the end of "others".

20. Page 3, line 76: Consider inserting the word "so" before "their potential".

21. Page 3, line 76: Add an "s" to the end of the word "remain".

22. Page 3, line 79: Add an "s" to the end of the word "account".

23. Page 3, line 84; Page 8, line 186: It's possible that the word "assess" should be used instead of "access".

24. Page 3, line 84: It's possible that the word "role" is intended instead of "rule".

25. Page 4, line 112; Page 5, line 143: Move "218" to after the word "world's".

26. Page 4, line 114; Page 5, line 117; Page 7, line 149: This may be a matter of style, but it's possible that the word "upper" is more precise in English than "superficial", which would be preferred in some other languages, such as French (example: superficiel).

27. Page 5, line 122: Replace "ed" at the end of "followed" with an "s".

28. Page 5, lines 123, 124, and 126: Delete the "s" at the end of the word "floodplains".

29. Page 5, line 129: Replace the "d" at the end of "build" with a "t".

30. Page 5, line 133: Delete the letters "es" from the end of the word "gases".

31. Page 5, line 134: Replace "are" with "is" after "forcing".

32. Page 5, line 137: Delete the "s" at the end of "differences".

33. Page 5, line 137: Change the uppercase "W" in "We" to a lowercase "w".

34. Page 5, line 138: Revise "allows to reduce" to "reduces".

35. Page 5, line 144: the word "total" in "the total land surface" may not be necessary. Figure 1 caption: Delete the "s" at the end of "behaviours".

36. Page 7, line 149: The word "globally" may not be needed.

37. Page 7, line 153: Revise "lowlands fraction" to "lowland fractions". Figure 2 caption: Revise "Floodplains behaviours" to "Floodplain behaviour" and add the letter "s" to the end of "month" in "(c) Number of month".

38. Page 8, lines 159 and 161: Insert "at a" before "maximum".

39. Page 8, line 183: Add an "s" to the end of the words "return" and "lead".

40. Page 8, line 184: Delete the "s" at the end of "mid-latitudes" and insert the word "the" before "impact of".

41. Page 8, line 186: Revise "discharges are" to "discharge is". Figure 3 caption: Delete the "s" at the end of the word "shows" in the sentence that begins "Bottom plots shows". Add an "s" to "correspond".

42. Page 9, line 197: Replacing "Whatever" with "Regardless" may increase clarity.

43. Page 9, line 198: Delete the "s" at the end of "floodplains".

44. Page 11, line 201: Replace the "s" at the end of "explains" with "ed".

45. Page 11, line 205: replace the word "of" with the word "for".

46. Page 11, line 214: Delete the "a" in "bay".

47. Page 11, line 215: Replace "in lesser extend" with "to a lesser extent". Add an "s" to the end of the word "Gange".

48. Page 11, line 216: Change "plane" to "flattening", "diminishing" or another word with a similar meaning that ends in "ing". Consider replacing "due to the strong flooded" with "due to strong flood".

49. Page 11, line 218: Delete the word "to" after "allow".

50. Page 11, line 219: The meaning of the words "to dawn" is not clear in this context.

51. Page 11, lines 220 to 225; Page 14, line 269: There is an extra space before a punctuation mark in these passages.

52. Page 11, line 227: Delete the letter "s" at the ends of the words "floodplains", "models", and "performances".

53. Page 12, line 235: Replace the "ed" at the end of "showed" with the letter "n".

54. Page 12, line 238: Add an "s" to the end of the word "lead".

55. Page 12, line 239: Delete the word "a" after "rise and".

56. Page 12, line 240: Change "in" after "transported" to "to". Delete the "s" at the end of "constitutes".

57. Page 12, Figure 5 subplot b title: Revise "Evapotranpiration" to "Evapotranspiration".

58. Page 13, line 243: Insert the word "in" between the words "increase" and "water".

59. Page 13, Figure 6 caption: Delete the "s" at the end of "grid-points". Add a "." at the end of the last sentence.

60. Page 14, line 266: Add an "s" at the end of the word "floodplain".

61. Page 14, line 270: Move the word "respectively" to the end of the sentence and delete the ")".

62. Page 14, line 276: Delete the word "an" after "leads to". Replace "enhancement" with "enhanced". Delete the word "of" before "evapotranspiration".

63. Page 14, line 288: Insert a ")" after "FLD".

64. Page 15, Figure 7 caption: Consider inserting a space between "percentage" and "point" in "percentagepoint".

65. Page 16, line 304: Insert a ")" after "season".

66. Page 16, line 310: The term "downward surface solar heat flux" is a bit confusing.

67. Page 16, line 322: replace "of" with "in" after "increase".

68. Page 18, line 328: Capitalize the first "a" in "amazonian".

69. Page 20, line 357: Add "'s" to the end of the word "atmosphere".

70. Page 20, line 363: Replace the "s" at the end of "biases" with a "d".

71. Page 20, line 367: Revise "underestimations of these" to "underestimation of".

72. Page 21, Figure 10 caption: Note that "2m" is written as "2-meter" some of the time.

73. Page 23, Figure 11 caption: Delete "s" at the end of "theirs".

74. Page 1: Consider capitalizing "S" in "simulated" for consistency with title case used throughout the rest of the title.

75. Page 2, line 20: Delete "continental".

76. Page 2, line 22: Delete "front of" and replace with "the face of".

77. Page 2, line 30: Delete the "s" at the end of "represents".

78. Page 2, line 32: Add an "s" to the end of "term".

79. Page 2, line 33: Delete the "s" at the end of "discharges".

80. Page 2, line 40: Depending on the intended meaning, it may be clearer to say "potential evaporation" rather than just "potential".

81. Page 2, line 47: The word "partitioned" may be more accurate that "distributed" here.

82. Page 3, line 53: Delete "the" after "North America, ".

83. Page 3, line 54: Replace "or" with "and" before "east Africa".

84. Page 3, line 54: Delete "these" before "water bodies".

85. Page 3, line 57: Delete "s" after "impacts".

86. Page 3, lines 57 and 58: Reverse the order of "enough documented" to read "documented enough".

87. Page 3, line 64: Delete "an" before "idealized".

88. Page 3, line 68: Delete the "s" at the end of "theirs".

89. Page 3, line 69: The word "capacity" may be preferred to "capability" in this context.

90. Page 3, line 70: The word "role" may be more appropriate than "rule" in this sentence.

91. Page 3, line 71: Consider including a comma after "rivers" to indicate that a phrase is being added to the main clause in this sentence.

92. Page 3, line 72: Delete the "s" at the end of "others".

93. Page 3, line 76: Consider inserting the word "so" before "their potential".

94. Page 3, line 76: Add an "s" to the end of the word "remain".

95. Page 3, line 79: Add an "s" to the end of the word "account".

96. Page 3, line 84; Page 8, line 186: It's possible that the word "assess" should be used instead of "access".

97. Page 3, line 84: It's possible that the word "role" is intended instead of "rule".

98. Page 4, line 112; Page 5, line 143: Move "218" to after the word "world's".

99. Page 4, line 114; Page 5, line 117; Page 7, line 149: This may be a matter of style, but it's possible that the word "upper" is more precise in English than "superficial", which would be preferred in some other languages, such as French (example: superficiel).

100. Page 5, line 122: Replace "ed" at the end of "followed" with an "s".

101. Page 5, lines 123, 124, and 126: Delete the "s" at the end of the word "floodplains".

102. Page 5, line 129: Replace the "d" at the end of "build" with a "t".

103. Page 5, line 133: Delete the letters "es" from the end of the word "gases".

104. Page 5, line 134: Replace "are" with "is" after "forcing".

105. Page 5, line 137: Delete the "s" at the end of "differences".

106. Page 5, line 137: Change the uppercase "W" in "We" to a lowercase "w".

107. Page 5, line 138: Revise "allows to reduce" to "reduces".

108. Page 5, line 144: the word "total" in "the total land surface" may not be necessary. Figure 1 caption: Delete the "s" at the end of "behaviours".

109. Page 7, line 149: The word "globally" may not be needed.

110. Page 7, line 153: Revise "lowlands fraction" to "lowland fractions". Figure 2 caption: Revise "Floodplains behaviours" to "Floodplain behaviour" and add the letter "s" to the end of "month" in "(c) Number of month".

111. Page 8, lines 159 and 161: Insert "at a" before "maximum".

112. Page 8, line 183: Add an "s" to the end of the words "return" and "lead".

113. Page 8, line 184: Delete the "s" at the end of "mid-latitudes" and insert the word "the" before "impact of".

114. Page 8, line 186: Revise "discharges are" to "discharge is". Figure 3 caption: Delete the "s" at the end of the word "shows" in the sentence that begins "Bottom plots shows". Add an "s" to "correspond".

115. Page 9, line 197: Replacing "Whatever" with "Regardless" may increase clarity.

116. Page 9, line 198: Delete the "s" at the end of "floodplains".

117. Page 11, line 201: Replace the "s" at the end of "explains" with "ed".

118. Page 11, line 205: replace the word "of" with the word "for".

119. Page 11, line 214: Delete the "a" in "bay".

120. Page 11, line 215: Replace "in lesser extend" with "to a lesser extent". Add an "s" to the end of the word "Gange".

121. Page 11, line 216: Change "plane" to "flattening", "diminishing" or another word with a similar meaning that ends in "ing". Consider replacing "due to the strong flooded" with "due to strong flood".

122. Page 11, line 218: Delete the word "to" after "allow".

123. Page 11, line 219: The meaning of the words "to dawn" is not clear in this context.

124. Page 11, lines 220 to 225; Page 14, line 269: There is an extra space before a punctuation mark in these passages.

125. Page 11, line 227: Delete the letter "s" at the ends of the words "floodplains", "models", and "performances".

126. Page 12, line 235: Replace the "ed" at the end of "showed" with the letter "n".

127. Page 12, line 238: Add an "s" to the end of the word "lead".

128. Page 12, line 239: Delete the word "a" after "rise and".

129. Page 12, line 240: Change "in" after "transported" to "to". Delete the "s" at the end of "constitutes".

130. Page 12, Figure 5 subplot b title: Revise "Evapotranpiration" to "Evapotranspiration".

131. Page 13, line 243: Insert the word "in" between the words "increase" and "water".

132. Page 13, Figure 6 caption: Delete the "s" at the end of "grid-points". Add a "." at the end of the last sentence.

133. Page 14, line 266: Add an "s" at the end of the word "floodplain".

134. Page 14, line 270: Move the word "respectively" to the end of the sentence and delete the ")".

135. Page 14, line 276: Delete the word "an" after "leads to". Replace "enhancement" with "enhanced". Delete the word "of" before "evapotranspiration".

136. Page 14, line 288: Insert a ")" after "FLD".

137. Page 15, Figure 7 caption: Consider inserting a space between "percentage" and "point" in "percentagepoint".

138. Page 16, line 304: Insert a ")" after "season".

139. Page 16, line 310: The term "downward surface solar heat flux" is a bit confusing.

140. Page 16, line 322: replace "of" with "in" after "increase".

141. Page 18, line 328: Capitalize the first "a" in "amazonian".

142. Page 20, line 357: Add "'s" to the end of the word "atmosphere".

143. Page 20, line 363: Replace the "s" at the end of "biases" with a "d".

144. Page 20, line 367: Revise "underestimations of these" to "underestimation of".

145. Page 21, Figure 10 caption: Note that "2m" is written as "2-meter" some of the time.

146. Page 23, Figure 11 caption: Delete "s" at the end of "theirs".

**Reply**:  We agree with the vast majority of your correction and modified the manuscript accordingly. Below, we only reply to the comments with which we disagree or for which we came up with a correction different from the one you suggested.

7. We wrote instead that " [evapotranspiration] reaches its potential rate".

11. We changed "these water bodies" into "inland water bodies".

12. & 13. We changed the whole sentence from " *The climate impacts of groundwater stored in large unconfined aquifers are however not yet enough documented while it acts as a lower boundary for the overlaying unsaturated soil through upward capillarity rise*" to : "*However, the way groundwater stored in large unconfined aquifers might impact climate worldwide has not yet been documented enough, even though these aquifers act as a lower boundary for the overlaying unsaturated soil it can affect through upward capillarity rise*".

21. Instead, we added a "s" to feedback.

23. Yes. And also page 8, line 166.

26. By "superficial", we meant the part of the soil column of the ISBA model, based on the Richard's equation (unsaturated soil). We understand what you mean but we are afraid "upper" might be misleading as it can refer to the whole unsaturated soil, and not just its upper part.
Line 114, we thought it was best to remove "superficial" from the sentence altogether.
Lines 115-119, we rewrote the 2 sentences to change the two occurrences of "superficial". Going from : "*The second difference compared to the standard version of the CNRM-CM6-1 climate model appears in the coupling between the aquifer water table and the superficial soil. While the standard coupling imposes the water table depth to be lower or equal to the hydrological*

*soil depth in ISBA to compute upward capillarity fluxes (Vergnes et al., 2014; Decharme et al., 2019), in our upgraded version the Richards equation is modified to allow the water table to penetrate into the superficial soil (Colin et al., 2023)."* to : *"The coupling between the aquifer simulated by CTRIP and the soil column simulated by ISBA has been modified compared to the standard version of the CNRM-CM6-1 climate model. While the standard coupling imposes the water table depth to be lower or equal to the hydrological soil depth in ISBA to compute upward capillarity fluxes (Vergnes et al., 2014; Decharme et al., 2019), in our upgraded version, the Richards equation is modified to allow the water table to penetrate into the soil column of ISBA (Colin et al., 2023)."*

Line 149, "superfical" was changed into "ISBA soil column".

Line 152, "Unsaturated superficial soil" was changed into "above unsaturated soil".

36. We changed "represent globally 12 %" into "amount to 12 %".

50. We rewrote this part of the sentence, which now is "*the strong underestimation of the simulated precipitation limits the development of floodplains*".

52. OK for "floodplain" and "performance", but we do mean to talk about land-atmosphere coupled models in general, hence the plural.

66. We changed it to "surface downward solar radiation". We also changed it in the caption of figure 8.

70. We don't think this would work. We do mean "the model's biases" as in "the biases of the model" ("les biais du modèle" in french).

72. Yes. We wrote "2m" in figures captions and in the data availability statement but felt it was better not to use this abbreviation in the text, where we prefered "2-meter". We think it is quite clear that 2m = 2-meter.
* * *
[Figure]

Figure 1: Schematic view of the CNRM-CM6-1 climate model in "AMIP mode", that is not coupled to the NEMO ocean model. The model couples the ARPEGE-Climat v6.3 general circulation model with the ISBA-CTRIP system to simulate atmospheric, land surface, and hydrological processes. The ISBA land surface model includes a one-layer vegetation interception scheme, $CO_2$-responsive plant transpiration, bare soil evaporation, snow sublimation, a 12-layer snow scheme, and a 14-layer explicit soil scheme for temperature, moisture, and soil ice. Soil moisture is computed within the rooting depth, which varies from 0.2 to 8 meters depending on the vegetation type, while soil temperature is simulated down to 12 meters. Subgrid hydrology accounts for surface runoff, deep drainage, and recharge fluxes, with groundwater interactions modeled over the low land fraction of the grid cell affected by the water table, $f_{wtd}$ (illustrated by the grey footprint around the river). The CTRIP hydrological model simulates river discharge using total runoff from ISBA, dynamically solving river velocity with Manning's formula and incorporating a flooding scheme to compute flood volume and extent. It also includes a two-dimensional groundwater scheme for unconfined aquifers, representing time variations in water table depth, interacting with rivers and fed by the recharge rate simulated by ISBA. The coupling between ISBA and CTRIP is achieved via the OASIS-MCT interface, allowing interactions between floodplains, soil, and atmosphere through evaporation, infiltration, and precipitation interception, as well as upward capillary fluxes into the unsaturated soil over $f_{wtd}$. Variables involved in these coupling processes are shown in black. Further details are available in Decharme et al. (2019).

---

## Author Response (AR2)

**Response to Editor decision's comments for manuscript egusphere-2024-3091**

**General Comment** — Both Reviewers are very positive on your submission. Following the edits that you have implemented and your replies to the Reviewer comments, I believe that your manuscript is suitable for publication in Earth System Dynamics. Before final acceptance, I would ask you to confirm whether the schematics in the new Fig. 1 have been taken from existing publications, or whether they have been created specifically for this paper. In the former case, you should explicitly reference the source(s) of the schematics that you used in the figure caption. Also consider whether the figure caption may be made more concise, and part of the information be included only in the main text. Finally, I would suggest editing l. 504 to read only "code availability", since you provide information on data availability in the previous paragraph.

**Reply**: Thank you very much. We confirm that the schematics in the new Fig. 1 have been created specifically for this paper. Thank you for your suggestion about the figure caption. However, we have opted to provide a detailed caption for Figure 1 to allow readers immediate access to essential information directly under the figure. We believe that this approach enhances the immediate understanding of the key components and interactions depicted, without requiring frequent reference to the main text. We believe this improves accessibility and clarity for the readers. Finally, "*Code and data availability*" has been changed to "*code availability*"